# The Pathogenic Yeast *Metschnikowia bicuspidata* var. *bicuspidata* in the Aquacultured Ecosystem and Its Biocontrol

**DOI:** 10.3390/jof9101024

**Published:** 2023-10-18

**Authors:** Khalef Hansali, Zhao-Rui Zhang, Guang-Lei Liu, Zhe Chi, Zhen-Ming Chi

**Affiliations:** 1College of Marine Life Sciences, Ocean University of China, Yushan Road, No. 5, Qingdao 266003, China; khalef@stu.ouc.edu.cn (K.H.); zrz@sina.com (Z.-R.Z.); liugl@ouc.edu.cn (G.-L.L.); cz1108@ouc.edu.cn (Z.C.); 2Laboratory for Marine Biology and Biotechnology, Qingdao National Laboratory for Marine Science and Technology, Qingdao 266003, China

**Keywords:** *M. bicuspidata* var. *bicuspidata*, the pathogenic yeast, milky disease, aquacultured animals, killer toxin, Massoia lactone

## Abstract

*M. bicuspidata* var. *bicuspidata* is a pathogenic yeast which can affect aquacultured and marine-cultured animals such as brine shrimp, ridgetail white prawn, chinook salmon, giant freshwater prawn, the Chinese mitten crab, marine crab, the mud crab, the mangrove land crab, the Chinese grass shrimp, sea urchins, sea urchins, *Daphnia dentifera* and even snails, causing a milky disease, and it has caused big economic losses in aquacultural and marine-cultural industries in the past. However, the detailed mechanisms and the reasons for the milky disease in the diseased aquatic animals are still completely unknown. So far, only some antimycotics, killer toxins and Massoia lactone haven been found to be able to actively control and kill its growth. The ecofriendly, green and renewable killer toxins and Massoia lactone have high potential for application in controlling the milky disease.

## 1. Introduction

*M. bicuspidata* belongs to one species of the genus *Metschnikowia* spp. and has four varieties: *M. bicuspidata* var. *bicuspidata, M. bicuspidata* var. *californica*, *M. bicuspidata* var. *chathamia* and *M. bicuspidata* var. *australis* [1,2]. All of them are the ascosporogenous yeasts with a single and acicular ascospore, their asci arise from budding cells, and reproduction of the ascosporogenous cells is generally by multilateral budding [3]. They can grow on glucose, galactose, maltose, lysine and 19% NaCl at 28 °C. In addition, the pathogenic yeast *M. bicuspidata* var. *bicuspidata* isolated from the diseased Chinese mitten crab in Panjin city, Liaoning Province, northwestern China, can grow well at 5 °C [4]. Yeasts in the genus *Metschnikowia* spp. are globally distributed in a variety of substrates including plants, insects, aquatic animals, freshwater and sea water in France, Romania, Russia, China, the United States, Canada and even Antarctic waters [5,6]. Recently, it has been well documented that *M. bicuspidata* var. *bicuspidata* is a pathogenic yeast which causes a milky disease in brine shrimp, ridgetail white prawn (*Exopalaemon carinicauda*), chinook salmon (*Oncorhynchus tshawytscha*), giant freshwater prawn (*Macrobrachium rosenbergii*), the Chinese mitten crab (*Eriocheir sinensis*), marine crab (*Portunus trituberculatus*), the mud crab (*Scylla paramamosain*), the mangrove land crab (*Ucides cordatus*), the Chinese grass shrimp (*Palaemonetes sinensis*), sea urchins, *Daphnia dentifera* and even snails. These animals can ingest *M. bicuspidata* var. *bicuspidata* ascospores or consume the diseased individuals which contain the pathogenic yeast cells, causing a yeast infection in these animals under the specific conditions of the high density aquaculture. Then, the spores in the host cavity begin to germinate to produce a large amount of hyphae and vegetative cells by budding, giving rise to the milky disease. In recent years, this milky disease has received great attention because it has led to big economic losses in aquaculture and maricultural industries in China, and the mortality rate of the diseased animals is over 20%. Unfortunately, it is still completely unclear how *M. bicuspidata* var. *bicuspidata* causes the aquacultured animals to have the milky disease and how to effectively biocontrol this disease. In this mini-review article, the milky disease in various aquacultured animals is summarized, potential causes of the milky disease are suggested, and biocontrol of the disease by using a killer toxin and Massoia lactone from liamocin is discussed.

## 2. Milky Disease Caused by the Pathogenic *M. bicuspidata* var. *bicuspidata*

In general, aquatic animals can be easily infected by pathogenic bacteria, viruses, filamentous fungi and protozoa. However, in recent years, it also has been found that *M. bicuspidata* var. *bicuspidata* is a pathogenic yeast that can cause the milky disease in many different economic aquatic animal species. It has been reported that nine strains of the *M. bicuspidata* var. *bicuspidata* were isolated from diseased brine shrimp (*Artemia salina*), which were grown in salt ponds containing 10% NaCl in Southern California [7]. It also has been reported that *Oncorhynchus tshawytscha* (chinook salmon) can be infected by *M. bicuspidata* var. *bicuspidata* from the infected *Artemia franciscana,* which was the feed for the chinook salmon, leading to mortality, systemic infection and necrosis of the diseased chinook salmon [8].

The pathogenic yeast *M. bicuspidata* var. *bicuspidata* also infected the Chinese swimming crab *Portunus trituberculatus,* the mud crab *Scylla paramamosain* and the mangrove land crab *Ucides cordatus.* For example, the marine crab *P. trituberculatus* was cultivated in293,000 Acres of seawater only in Zhejiang Province, China. During the period of 2002 and 2003, a total production of 39,000 t of the cultivated marine crab was obtained and they cost CNY 2.028 billion (about USD 0.2897 billion) at that time. However, 30–50% of the cultivated marine crab had the milky disease caused by *M. bicuspidata* var. *bicuspidata*, causing huge economic losses in this area during this period [9]. The purified yeast strain *M. bicuspidata* var. *bicuspidata* from the diseased parts of the marine crab could develop the same milky symptoms, such as crab emaciation, opaque muscles, a large amount of yeast cells in the muscle, heart and hepatopancreas and a smell of the fermented food by yeasts as the native type strain [10] (Figure 1A).

*M. bicuspidata* var. *bicuspidata* (the major cause of yeast infections) together with *Saccharomyces cerevisiae* and *Candida albicans* could also infect 6- to 11-month-old giant freshwater prawn *Macrobrachium rosenbergii* at water temperatures below 17 °C [11]. The pathogen isolated from the Chinese mitten crab with the milky disease in Panjin, Liaoning Province, China, was also the yeast *M. bicuspidata* var. *bicuspidata* [5], and the mortality rate of the Chinese mitten crab by this yeast was over 20%. Very recently, it has been well documented that there was a new milky disease outbreak in ridgetail white prawn (*Exopalaemon carinicauda*) in coastal areas of Jiangsu Province, China [12].

The milky disease of the giant freshwater prawn caused by *M. bicuspidata*, var. *bicuspidata* included milky hemolymph, a yellow exoskeleton, opaque muscles, and a swollen hepatopancreas. Histopathological research of the diseased giant freshwater prawn showed that there were marked edema and extensive necrotic lesions associated with a high yeast cell density and inflammation within the muscles, hepatopancreas and other internal organs such as the heart, ovary and intestine. Yeast cell numbers isolated from various tissues of the diseased giant freshwater prawn ranged from 4.5 × 10^8^ to 9.0 × 10^9^ colony forming units per 100 mg of each diseased tissue. Because there were very high concentrations of the pathogenic yeast cells in the hemolymph of the naturally infected prawns, they could easily be spread through the circulatory system to reach various parts of the diseased animals [11].

Clinical signs of the milky disease in the Chinese mitten crab, *Eriocheir sinensis,* caused by *M. bicuspidata* var. *bicuspidata*, a pathogenic yeast found in aquatic invertebrates, include a milky hemolymph, weak responses to stimulants, swollen pereopod and claw joints and opaque or whitish muscles, culminating in tissue or organ failure and death (Figure 1B). Histopathological analysis of the diseased crabs showed a marked disorder of fibers, discrete necrotic lesions and high yeast cell density in the muscles, heart, gills and other organs [5]. The purified yeast cells can grow well on solid YPD plates at 28 °C and 15 °C within two days or three days. Even the purified yeast cells can grow at 5 °C within 15 days. The sequences of ITS and 26S rDNA showed that the purified pathogenic yeast belongs to one member of *M. bicuspidate* var. *bicuspidata* [4].

The diseased ridgetail white prawn infected by *M. bicuspidata* var. *bicuspidata* MQ2101 exhibited typical symptoms, such as a reddish body color, whitish muscles, enlarged appendages, slow swimming, large amounts of the milky yeast cells in the body, liquefied muscles, opaque white color and milky white hemolymph [12].

However, the detailed cause and mechanisms of the milky disease caused by *M. bicuspidata* var. *bicuspidata* have been completely unknown to date.

## 3. Possible Causes of the Milky Disease

Recently, many researchers have tried to elucidate the possible causes of the milky disease by sequencing and annotating the genomes of the pathogenic *M. bicuspidata* var. *bicuspidata* strains isolated from the diseased aquatic animals. For example, the size of the whole genome of *M. bicuspidata* var. *bicuspidata* MQ2101 isolated from diseased *E. carinicauda* is 15.98 Mb, and its genome contains 3934 coding genes. The genome has been predicted to contain 484 CAZyme-encoding gene homologues encoding glycoside hydrolases, carbohydrate esterases, glycosyl transferases and auxiliary activities and the genes encoding amino acid transporters, aspartyl protease, cytochrome P450, ABC transporter, ferric reductase, multicopper oxidase, glycosyltransferase and lysophospholipase. Especially, we think that secreted lipases, lysophospholipases and aspartyl proteases may be involved in the pathogenic processes of the yeast because they may hydrolyze lipids and proteins in the diseased animals and use the hydrolysis products for heavy yeast cell growth, the so-called milky disease [4,12,13,14,15].

However, the size of the genome of *M. bicuspidata* var. *bicuspidata* LNES0119 pathogenic for the Chinese mitten crab *E. sinensis* was 16.13 Mb and its genome encodes 5567 putative predicted genes, among which 30 genes were thought to be related to its pathogenicity [6]. Especially, secreted eukaryotic aspartyl protease, lipase, phospholipases, serine carboxypeptidase, carbohydrate-active enzymes, hyphally regulated cell wall protein N-terminal and subtilase may be involved in its pathogenicity because the same pathogenic yeast isolated from the diseased Chinese mitten crab *E. sinensis* was confirmed to be able to grow in the extracts of the muscle, gill, heart tissues and intestinal tracts of the healthy Chinese mitten crabs by using their reducing sugars, amino acids and other nutrients [4]. Indeed, it was also found that lipases, proteases and phospholipases are the virulence factors that increase the pathogenicity of *Candida albicans,* a common opportunistic pathogen in the human body [16,17].

We found that the same pathogenic yeast *M. bicuspidata* var. *bicuspidata* isolated from the diseased Chinese mitten crab in Panjin city, Liaoning Province, northwestern China, can also grow well at 5 °C as mentioned above [4], meaning that the pathogenic yeast *M. bicuspidata* var. *bicuspidata* is a psychrotolerant yeast, which exhibits optimum growth temperatures of about 20–30 °C and a minimum growth temperature of 0 °C [18]. That is why it can infect the Chinese mitten crab even under the ice during the winter of Panjin city in the Northeast China, and the infection rate of the adult crabs in winter was higher than that in summer [19]. Indeed, the same yeast species *M*. *bicuspidata* var. *australis* strain UFMG-CMY6158 isolated from a marine macroalgae collected in Admiralty Bay of King George Island in Keller Peninsula, Antarctica, and *M. bicuspidata* var. *australis* W7-5 obtained from the sea mud of South bay of Fildes Peninsula in Antarctica can also grow well at 5 °C [2,18].

Moore et al. [8] proposed that *M. bicuspidata* var. *bicuspidata* actively infested the intestine of salmon by releasing needle-like ascospores via dehiscence, and the needle-like ascospore has the ability to penetrate inner layers of the host intestine, which permits the pathogenic yeast to multiply in the intestine and makes the grown yeast cells eventually be distributed throughout the whole body.

We think that the highly intensive cultivation of aquatic animals, the deterioration of their surroundings and the reduced immunity of aquatic animals may also allow the pathogenic yeast *M. bicuspidata* var. *bicuspidata* to invade the cultured animals and heavily grow in their bodies so that the milky disease occurs. Indeed, it has been well known that the pathogenic yeast *Candida albicans* can easily infect the immunocompromised patients, such as HIV-infected individuals, elderly people, cancer patients and so on [20].

However, there is still no genetic evidence to demonstrate which genes are closely related to their pathogenicity. Therefore, the relevant genes in the genomic DNA in the pathogenic yeast must be cloned, characterized, deleted and overexpressed. Then, the healthy aquatic animals must be challenged with the obtained various mutants in order to confirm which genes are closely related to the pathogenicity of the pathogenic *M. bicuspidata* var. *bicuspidata*. At the same time, the close relationship between the enzyme activities and their pathogenicity must be resolved. Fortunately, the whole genome of the psychrophilic yeast *M. bicuspidata* var. *australis* W7-5, a variety from the same species isolated from Antarctica, has been successfully edited and genetically modified using a specific and highly efficient Cre/loxP system developed in this laboratory. All the molecular techniques, including construction of a plasmid, transformation, removal of the antibiotic resistant genes and deletion, overexpression and complementation of encoding genes for understanding the functions of the genome of the pathogenic *M. bicuspidata* var. *bicuspidata* can be referred [18,21].

## 4. Biocontrol of the Pathogenic *M. bicuspidata*, var. *bicuspidata*

Although the antibiotics such as ketoconazole, fluconazole, econazole, clotrimazole, amphotericin B, itraconazole and nystatin have inhibitory activity against *M. bicuspidata* var. *bicuspidata* 2EJM001 isolated from farmed *E. sinensis* with milky disease [22], the absorption, distribution, metabolism and drug residue of the antibiotics should be considered when they are used for clinical treatment. In recent years, antibiotics have been thought to be an environmental pollutant that causes the wide spread of antibiotics-resistant microorganisms and antibiotics-resistant genes [23]. Most importantly, in China, all the antibiotics are prohibited to be used in the aquaculture system in order to stop antibiotics pollution and the spread of the antibiotic resistance gene and bacteria. Therefore, these drugs can only be used for emergency treatment and should not be used for routine prevention and treatment of the milky disease in the aquaculture ecosystem.

It was also found that nystatin, benzalkonium bromide and extract of goldthread root and garlic are active against the yeast *M. bicuspidata* var. *bicuspidata* WCY pathogenic for marine crab. However, the compounds with MIC (the minimum inhibitory concentration) are harmful to the cultivated crab in marine environments and it is completely impossible to use such expensive antibiotics mentioned above to control the milky disease of a marine crab in the open sea [9,10].

Therefore, it is very significant to find the possibility of using alternate active bioproducts produced by fungal strains and any other microorganisms for inhibition and killing of the yeast *M. bicuspidata* var. *bicuspidata*. Furthermore, these active bioproducts produced by fungal strains and any other microorganisms must be eco-friendly, green and renewable. So far, only the killer toxin and Massoia lactone have been found to be such active bioproducts against the pathogenic yeast *M. bicuspidata* var. *bicuspidata*.

### 4.1. Biocontrol of the Pathogenic Yeast M. bicuspidata var. bicuspidata Using Killer Toxins

Killer toxins synthesized and secreted by some yeast strains are the proteins that destroy sensitive cells of the same or related yeast genera including some pathogenic yeasts via a two-step mode of action [24]. Such killer toxins may also be used to kill the pathogenic yeast *M. bicuspidata* var. *bicuspidata* mentioned above and treat the milky disease outbreak in the aquaculture system. Indeed, aquatic *Metschnikowia* species (*M. bicuspidata* var. *australis, M. bicuspidata* var. *bicuspidata, M. bicuspidata* var. *krissii* and *M. bicuspidata* var. *zobellii*) were found to be sensitive to 10 out of 12 antimycotics secreted by *Pichia membranifaciens* strains [25].

After screening of more than 300 yeast strains from different sources in marine environments, it was found that *Pichia anomala* strain YF07b and *Williopsis saturnus* WC91-2 had the highest ability to produce a killer toxin against the pathogenic yeast *M. bicuspidata* var. *bicuspidata* WCY in the marine crabs [10,24]. The deduced protein from the gene encoding the *P. anomala* YF07b killer toxin has 427 amino acids [24]. The molecular weight of the purified killer toxin is 47.0 kD and the optimal pH and temperature of the purified killer toxin for killing the pathogenic yeast *M. bicuspidata* var. *bicuspidata* WCY are 4.5 and 40 °C, respectively. A large amount of monosaccharides and disaccharides can be released from larminarin β-1, 3-D-glucan after the hydrolysis of the larminarin with the purified killer toxin, indicating the purified killer toxin has a high exo-β-1, 3-D-glucanase activity against β-1, 3-D-glucan in the cell wall of the pathogenic yeast. The purified killer toxin also actively killed the whole cells of the pathogenic yeast isolated from the marine crab [24]. Meanwhile, the molecular mass of the purified killer toxin produced by *W. saturnus* WC91-2 is 11.0 kDa and it has high killing activity against the pathogenic yeast *M. bicuspidata* var. *bicuspidata* WCY, too, but could not hydrolyze laminarin. The optimal conditions for action of the purified killer toxin against the pathogenic yeast *M. bicuspidata* var. *bicuspidata* WCY were 10% NaCl, pH 3–3.5 and temperature 16 °C. The gene encoding the killer toxin from the marine killer yeast WC91-2 has 378 bp and the deduced protein from the cloned gene encoding the killer toxin has 125 amino acids with a calculated molecular weight of 11.6 kDa, which is similar to that of the purified killer toxin from *W. saturnus* WC91-2 [24]. The killer toxin produced by *W. saturnus* WC91-2 acts on the cell wall of the sensitive yeast cells by interfering with the synthesis of β-1,3-D-glucan of the sensitive cell wall, leading to defective integrity of the cell wall and cell death of *M. bicuspidata* var. *bicuspidata* WCY [26].

Furthermore, it is interesting to have observed that the cold-active killer toxin (its optimal temperature is 15 °C) produced by a psychrotolerant yeast *Mrakia frigida* 2E00797 isolated from sea sediment in Antarctica could also actively kill the pathogenic yeast *M. bicuspidata* var. *bicuspidata* WCY [24]. In addition to the pathogenic *M. bicuspidata* var. *bicuspidata* WCY in the marine crab, the killer toxin produced by *M. frigida* 2E00797 could also kill *Candida tropicalis* and *C. albicans,* which are also common opportunistic fungal pathogens in human and animals [20]. The molecular weight of the purified *M. frigida* 2E00797 killer toxin is 55.6 kDa, but the gene encoding the killer toxin still has not been unidentified. The purified *M. frigida* 2E00797 killer toxin was able to actively kill the whole cells of *M. bicuspidata* var. *bicuspidata* WCY (Figure 2A)*, C. tropicalis* and *C. albicans* at 15 °C, but could not kill the protoplast of the sensitive yeasts. This indicates that the action site of the *M. frigida* 2E00797 killer toxin is the cell wall of the pathogenic *M. bicuspidata* var. *bicuspidata* WCY, not its plasma membrane. Indeed, many data have shown that cell walls, especially β-1,3 and β-1,6-D-glucan in the yeast cell walls, are the primary receptors for most killer toxin activity [24]. That is why the action site of the *M. frigida* 2E00797 killer toxin, the *W. saturnus* WC91-2 killer toxin and the *P. anomala* strain YF07b killer toxin is the cell wall of the pathogenic *M. bicuspidata* var. *bicuspidata* WCY, and the *P. anomala* strain YF07b killer toxin can hydrolyze larminarin. This means that most of the killer toxins can be specifically targeted towards the cell wall of the pathogenic *M. bicuspidata* var. *bicuspidata* WCY, not towards to cells of the cultivated animals. Therefore, when the killer toxins are used to cure the milky disease, they are not toxic to the cultured animals and are safe and eco-friendly.

In addition, many other marine yeasts including *Wickerhamomyces anomalus* HN1-2, isolated from the mangrove ecosystem and *Kluyveromyces siamensis* HN12-1 isolated from a mangrove ecosystem. *Pichia guilliermondii* GZ1, *Debaryomyces hansenii* hcx-1 and *Aureobasidium pullulans* HN2.3 also produce killer toxins against the pathogenic yeast *M. bicuspidata* var. *bicuspidata* WCY [24].

### 4.2. Biocontrol of the Pathogenic Yeast M. bicuspidata var. bicuspidata Using Massoia Lactone Released from Liamocin

Liamocins, the secondary metabolites synthesized and secreted by *Aureobasidium melanogenum* and *A. pullulans* consist of a single mannitol or a single arabitol head group and three or four 3,5-dihydroxydecanoic ester tail groups, or the head groups are directly esterified with three or four 3,5-dihydroxydecanoic ester tails [27,28] (Figure 3). The liamocin titers synthesized by the engineered strains of *A. melanogenum* have reached over 60.1 g/L [29,30]. Under alkaline condition, liamocins can be chemically hydrolyzed to release 3,5-dihydroxydecanoic acid, which can be converted to Massoia lactone under acidic conditions [27] (Figure 3). Therefore, Massoia lactone obtained in these studies is an α, β,-unsaturated d-lactone moiety substituted at the C6 position by an alkyl chain of five carbons and designated as C-10 Massoia lactones (Figure 3). The obtained Massoia lactone is very stable and highly active against many fungal pathogens including the pathogenic crop fungi, the pathogenic fungi on human and animal skin and the pathogenic yeast *M. bicuspidata* WCY in the marine crab [29,31].

Recently, we have found that both free Massoia lactone released from liamocins produced by *A. melanogenum* and the same Massoia lactone loaded in the nanoemulsions have high bioactivity against the pathogenic yeast strain from the diseased Chinese mitten crab *E. sinensis* (Figure 2B) and the diseased marine crabs. The minimal inhibitory concentrations and the minimal yeast-cidal concentration in the liquid culture against the pathogenic yeast strains were 0.15 mg/mL and 0.34 mg/mL, respectively. Furthermore, both free Massoia lactone and Massoia lactone loaded in the nanoemulsions could damage the cell membrane, even break the whole cells of the pathogenic yeast strains and cause cellular necrosis of the pathogenic yeasts. Therefore, both free Massoia lactone and Massoia lactone loaded in the nanoemulsions could be used to effectively kill the pathogenic yeast strains in an aquatic system and effectively control the milky disease in the Chinese mitten crab and the marine crab. In addition, Massoia lactone loaded in the nanoemulsions also had DPPH radical scavenging activity and hydroxyl radical scavenging activity [32]. However, the insolubility and weak volatility of the free Massoia lactone seriously limit its applications in aquaculture industries. So, various delivery systems using nanotechnology for improving the stability, solubility, high activity and bioavailability of Massoia lactone are necessary. Therefore, Massoia lactone loaded in the nanoemulsions may be more easily used for improving the delivery and dispersal of Massoia lactone in aquacultural and marine-cultural systems than free Massoia lactone [32]. In addition, the sensitive yeast cells have multiple sites acted on by Massoia lactone, so that unlike the antibiotics, it will be difficult for the pathogenic yeasts to develop to be resistant to the action of Massoia lactone [27].

The commercial C-10 Massoia lactone is now being extracted from the aromatic bark of *Cryptocarya massoy*, a Massoia tree which is of great ecological importance in tropical and subtropical regions like in Indonesia. However, because the Massoia tree only contains a small amount of Massoia lactone, it is very expensive and is not practical to obtain Massoia lactone in a large scale from the aromatic bark of *C. massoy* [27]. Another way to obtain Massoia lactone is chemical synthesis using hexanoyl chloride and ethyl acetoacetate as the substrates. During the chemical reactions, many toxic and expensive compounds and chemical catalysts must be used, the temperatures (190 °C, 130 °C and 80 °C) of some reactions are very high, and the process is very complex. So, the chemical way is not green and ecofriendly to environments and is not welcomed by modern society [33]. The best way to obtain Massoia lactone is to make liamocin chemically release 3,5-dihydroxydecanoic acid, which can be transformed into Massoia lactone under acidic conditions as mentioned above [27] (Figure 3), because a large amount of liamocin (over 60 g/L) can be produced by the engineered yeast-like fungal strain via fermentation on a large scale [28,29].

## 5. Conclusions

It has been strongly confirmed that the milky disease in some animals of freshwater and seawater are caused by the pathogenic yeast *M. bicuspidata* var. *bicuspidata*, the ascosporogenous yeast and the mortality rates of the infected animals were over 20%. Some proteinases and lipases and other hydrolytic enzymes synthesized and secreted by the pathogenic yeast *M. bicuspidata* var. *bicuspidata* may hydrolyze the proteins and lipids in the affected animals and use the hydrolysis products as carbon and nitrogen sources for the heavy yeast cell growth (milky symptoms) in different parts of the dead body. However, the real reasons for the milky disease are still unclear and are awaiting to be explored. Therefore, it is necessary to obtain enough evidence to clarify whether the proteinases and lipases and other hydrolytic enzymes are responsible for the milky disease through whole genome editing and biosynthetic biology in the pathogenic yeast and how the proteinases, lipases and other hydrolytic enzymes make the milky disease. Although killer toxins and Massoia lactone can actively kill the pathogenic yeast *M. bicuspidata* var. *bicuspidata*, it is still unknown how to practically apply the killer toxins and Massoia lactone to control the milky disease of cultured animals in freshwater and seawater systems.

## Figures and Tables

**Figure 1 jof-09-01024-f001:**
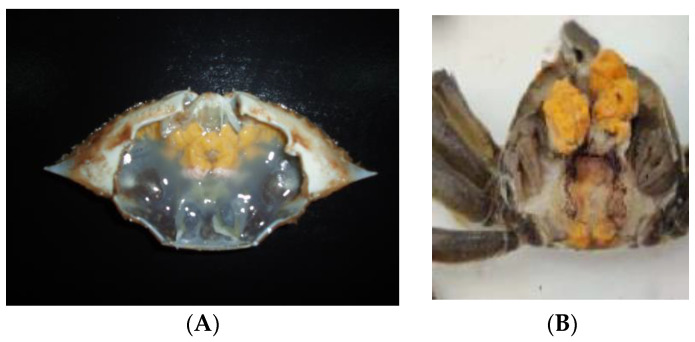
The pathogenic yeast-infected marine crab (**A**) and the Chinese mitten crab (**B**).

**Figure 2 jof-09-01024-f002:**
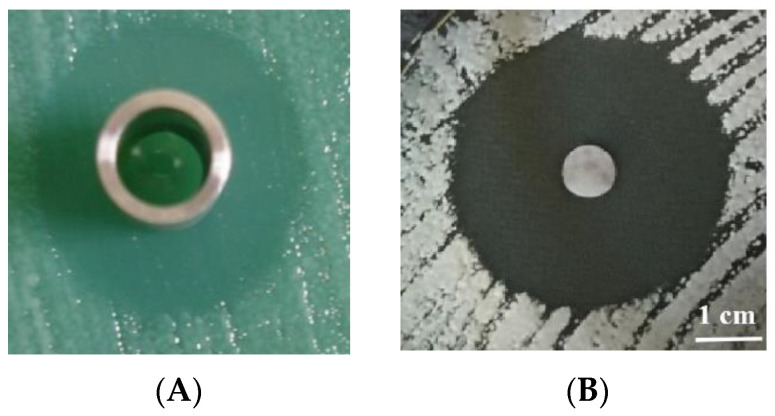
Clear zones (**A**) of the pathogenic *M. bicuspidata* var. *bicuspidata* WCY formed by the cold active killer toxin produced by *M. frigida* 2E00797 and that (**B**) of *M. bicuspidata* var. *bicuspidata* WCY pathogenic for the Chinese mitten crab formed by Massioa lactone.

**Figure 3 jof-09-01024-f003:**
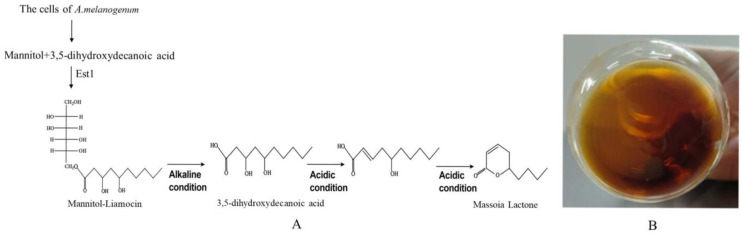
Massoia lactone released from liamocin produced by *A. melanogenum* (**A**) and the appearance of prepared Massoia lactone (**B**).

## Data Availability

Not applicable.

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
