# Peer review of "The Pathogenic Yeast Metschnikowia bicuspidata var. bicuspidata in the Aquacultured Ecosystem and Its Biocontrol"

_jof, 2023, doi:10.3390/jof9101024_

Round 1

Reviewer 1 Report

The Authors present an interesting work dealing with Metschnikowia bicuspidata var. bicuspidata, a yeast responsible for disease in aquacultured and marine cultured animals,  still presenting a lot of unknown aspects for what concerns pathogenesis, treatment and control.

In my opinion, the manuscript could be of interest for the readers of Journal of Fungi after some modifications, which are expressed here below:

-the running title should be changed, in order to immediately focus the topic of the work

- the Authors should check with particular attention the whole document and uniform characters, font and colors, according to the Author's instructions

-the reference section should be revised, avoiding the double numeration

- all Latin names should be in Italics

In addition, I would recommend an extensive revision of the scientific English used in the manuscript, in order to improve its clearness.

The Authors present an interesting work dealing with Metschnikowia bicuspidata var. bicuspidata, a yeast responsible for disease in aquacultured and marine cultured animals,  still presenting a lot of unknown aspects for what concerns pathogenesis, treatment and control.

In my opinion, the manuscript presents a series of points of interest for the readers of Journal of Fungi and could deserve publication after some modifications, which are expressed here below:

-the running title should be changed, in order to immediately focus the topic of the work

- the Authors should check with particular attention the whole document and uniform characters, font and colors, according to the Author's instructions

-the reference section should be revised, avoiding the double numeration

- all Latin names should be in Italics

In addition, I would recommend an extensive revision of the scientific English used in the manuscript, in order to improve its clearness.

Author Response

The Authors present an interesting work dealing with Metschnikowia bicuspidata var. bicuspidata, a yeast responsible for disease in aquacultured and marine cultured animals, still presenting a lot of unknown aspects for what concerns pathogenesis, treatment and control.

Thank you very much for your kind comments.

In my opinion, the manuscript could be of interest for the readers of Journal of Fungi after some modifications, which are expressed here below:

-the running title should be changed, in order to immediately focus the topic of the work

The yeast pathogen and its biocontrol

- the Authors should check with particular attention the whole document and uniform characters, font and colors, according to the Author's instructions

We have done these.

-the reference section should be revised, avoiding the double numeration

Ok, We have done these.

- all Latin names should be in Italics

All Latin names have been in Italics

In addition, I would recommend an extensive revision of the scientific English used in the manuscript, in order to improve its clearness.

The extensive revision of the scientific English used in the manuscript has been done.

Comments on the Quality of English Language

The Authors present an interesting work dealing with Metschnikowia bicuspidata var. bicuspidata, a yeast responsible for disease in aquacultured and marine cultured animals,  still presenting a lot of unknown aspects for what concerns pathogenesis, treatment and control.

You are right.

In my opinion, the manuscript presents a series of points of interest for the readers of Journal of Fungi and could deserve publication after some modifications, which are expressed here below:

Thank you for your kind comments.

-the running title should be changed, in order to immediately focus the topic of the work

The yeast pathogen and its biocontrol

- the Authors should check with particular attention the whole document and uniform characters, font and colors, according to the Author's instructions

Ok. I have checked uniform characters, font and colors, according to the Author's instructions

-the reference section should be revised, avoiding the double numeration

- all Latin names should be in Italics

The reference section has been revised based on your comments.

In addition, I would recommend an extensive revision of the scientific English used in the manuscript, in order to improve its clearness.

The extensive revision of the scientific English in the manuscript has been done.

Submission Date

07 September 2023

Date of this review

26 Sep 2023 11:46:50

Reviewer 2 Report

Dear authors,

The manuscript “The pathogenic yeast Metschnikowia bicuspidata var. bicuspidata in the aquacultured ecosystem and its biocontrol” is an interesting mini review about an important milky disease in marine organisms with economic value. Title is according to the thematic is concise, specific, and relevant as indicate the authors guide.

In the word attached file you can found some request and suggestions.

Best regards

Author Response

All Latin names should be in Italics

All Latin names have been in Italics

In addition, I would recommend an extensive revision of the scientific English used in the manuscript, in order to improve its clearness.

The extensive revision of the scientific English used in the manuscript has been done.

Comments on the Quality of English Language

The Authors present an interesting work dealing with Metschnikowia bicuspidata var. bicuspidata, a yeast responsible for disease in aquacultured and marine cultured animals,  still presenting a lot of unknown aspects for what concerns pathogenesis, treatment and control.

You are right.

In my opinion, the manuscript presents a series of points of interest for the readers of Journal of Fungi and could deserve publication after some modifications, which are expressed here below:

Thank you for your kind comments.

-the running title should be changed, in order to immediately focus the topic of the work

The yeast pathogen and its biocontrol

- the Authors should check with particular attention the whole document and uniform characters, font and colors, according to the Author's instructions

Ok. I have checked uniform characters, font and colors, according to the Author's instructions

-the reference section should be revised, avoiding the double numeration

- all Latin names should be in Italics

The reference section has been revised based on your comments.

In addition, I would recommend an extensive revision of the scientific English used in the manuscript, in order to improve its clearness.

The extensive revision of the scientific English in the manuscript has been done.

Submission Date

07 September 2023

Date of this review

26 Sep 2023 11:46:50

Observations and suggestions to the author´s

Abstract 128 words is correct.

Ok

The text has a little less than 4000 words as authors guide indicate.

This is a mini review article.

Line 24. Please change the word antibiotics to antimycotics.

“antibiotics” has been changed to “antimycotics”.

Line 15. Please add zip code, 266003 as indicate in the author´s guide.

266003 has been added.

Line 16. Please add Yushan Road, No. 5.

Yushan Road, No. 5 has been added.

Line 19. As a suggestion author could infantized an important date about the loses

for this disease.

“in the past” has been added.

Line 21. Authors can add sea urchins, Daphnia dentifera and even snails to

expand the yeast host.

“sea urchins, Daphnia dentifera and even snails” has been added.

Line 39-41. Please use italic typography in scientific names.

Ok, we have done these.

Line 40. Please review index fungorun page, in this are registered 5 varieties.

https://www.indexfungorum.org/Names/Names.asp

Metschnikowia bicuspidata var. bicuspidate has been changed to M. bicuspidata var. bicuspidata

Line 41. Please chanege catifornica for californica.

“catifornica” has been changed to “californica”.

Line 42. As a suggest change the ascospore to ascosporogenous yeast.

“The ascospore” has been changed to “ascosporogenous yeast”.

Line 43. Author referred to vegetative cell instead vegetable cells.

“vegetable cells” has been changed to “vegetative cells”

Line 52-57. Please use italic typography in scientific names.

We have done these.

Line 57. Change var bicuspidate to var. bicuspidata.

var bicuspidate has been changed to var. bicuspidata

Line 59. Which are the specific conditions to cause a disease.

“the high density aquaculture” has been added.

Line 64. Please use italic typography in scientific names.

Italic typography in scientific names has been used.

Line 69. Change M. bicuspidate var. Bicuspidate to M. bicuspidate var. bicuspidata.

  1. bicuspidate var. Bicuspidate has been changed to M. bicuspidata var. bicuspidata

Line 71. Please use the same typography size in Protozoon, il looks biggest

OK.

Line 72. Please change var. bicuspidate by var. bicuspidata.

“var. bicuspidate” has been changed to “var. bicuspidata”.

Line 74. Please change var. bicuspidate by var. bicuspidata.

“var. bicuspidate” has been changed to “var. bicuspidata”.

Line 75. Please use S in Southern.

Ok. southern has been changed to Southern.

Line 77. Please delete comma in M. bicuspidata.

The “,” has been deleted.

Line 83. Please check the size typography in acres, it looks big. Use the international system units to acres as indicate authors guide.

Ok, Acres

Line 85. Could you please change the value of 2028 billions of Chinese currencies

to dollars, this currency will allow to the reader a quicker idea of the value of these

economic losses.

(about 0.2897 billions of US dollars)

Line 88. Please delete comma in M. bicuspidata.

“,” has been deleted.

Line 99. Please change var. bicuspidate by var. bicuspidata.

var. bicuspidate had been changed to var. bicuspidata

Line. 115. Please change var. bicuspidate by var. bicuspidata.

var. bicuspidate has been changed to var. bicuspidata.

Line 117. Please replace Fig B by Figure B. as author´s guide indicate.

Fig B has been changed to Figure B

Line 123. Please change M. bicuspidate var. bicuspidate by M. bicuspidate var.

bicuspidata.

  1. bicuspidate var. bicuspidate has been changed to M. bicuspidata var. bicuspidata.

Line 124. Please change var. bicuspidate by var. bicuspidata.

  1. bicuspidate var. bicuspidate has been changed to M. bicuspidata var. bicuspidata.

Line 129. Please change var. bicuspidate by var. bicuspidata.

  1. bicuspidate var. bicuspidate has been changed to M. bicuspidata var. bicuspidata.

Line 133. Please change var. bicuspidate by var. bicuspidata.

  1. bicuspidate var. bicuspidate has been changed to M. bicuspidata var. bicuspidata.

Line 134. Please change var. bicuspidate by var. bicuspidata.

  1. bicuspidate var. bicuspidate has been changed to M. bicuspidata var. bicuspidata.

Line 144. Please change var. bicuspidate by var. bicuspidata.

  1. bicuspidate var. bicuspidate has been changed to M. bicuspidata var. bicuspidata.

Line 154 and 155. Please use the same size typography in opportunistic pathogen

in human body.

We have done this.

Line 159. Please change var. bicuspidate by var. bicuspidata and use the same size typography in Psychrotolerant yeast which exhibits.

  1. bicuspidate var. bicuspidate has been changed to M. bicuspidata var. bicuspidata.

Line 160 Include the initial square bracket in reference 18

Ok, [18]

Line 168. Please change var. bicuspidate by var. bicuspidata.

  1. bicuspidate var. bicuspidate has been changed to M. bicuspidata var. bicuspidata.

Line 173. Please use the same size typography in deterioration.

We have done this.

Line 175. Please change var. bicuspidate by var. bicuspidata.

  1. bicuspidate var. bicuspidate has been changed to M. bicuspidata var. bicuspidata.

Line 177-178. Please use the same size typography in immunocompromised

patients, such as HIV -infected individuals, elder people, cancer patients and so on

20.

Ok, we have done these.

Line 184. Please change var. bicuspidate by var. bicuspidata.

  1. bicuspidate var. bicuspidate has been changed to M. bicuspidata var. bicuspidata.

Line 188-190. These lines created a little confusion for me, I consider that it could

be said that techniques for understanding the genome of M. bicuspidate var.

bicuspidate are referred to in 18, 21, since the work to develop, these techniques was done using M. biscuspidata var australis.

All the molecular techniques, including construction of plasmid, transformation, removal of the antibiotic resistance gene and deletion, overexpression and complementation of the functional genes for understanding functions of the genome of the pathogenic M. bicuspidata var. bicuspidata can be referred [18, 21].

Line 190. Please change M. bicuspidate var. bicuspidate by M. bicuspidata var.

bicuspidata.

  1. bicuspidate var. bicuspidate has been changed to M. bicuspidata var. bicuspidata.

Line 191. Please change M. bicuspidate var. bicuspidate by M. bicuspidata var.

bicuspidata.

  1. bicuspidate var. bicuspidate has been changed to M. bicuspidata var. bicuspidata.

Line 192. I considered more appropriated the term antimycotic.

You are right. But the name antibiotics are being used very common

Line 194. Please change var. bicuspidate to var. bicuspidata.

  1. bicuspidate var. bicuspidate has been changed to M. bicuspidata var. bicuspidata.

Line 198. Please delete of in most of

“In Most of” has been deleted.

Line 204. Please change var. bicuspidate to bicuspidata.

  1. bicuspidate var. bicuspidate has been changed to M. bicuspidata var. bicuspidata.

Line 210. Why do the authors mention that alternative products for the control of

milky disease must be derived from fungi? Maybe it is possible found alternatives

in other sources like microbials, vegetables or animals as masoia lactone are

derived from bark of Cryptocarya spp. mainly from C. massoia, other species and

other plants and from fungi as Aureobasidium, Aspergillus among others.

“and any other microorganisms” has been added.

Line 211. Please change var. bicuspidate to bicuspidata.

  1. bicuspidate var. bicuspidate has been changed to M. bicuspidata var. bicuspidata.

Line 214. Please change var. bicuspidate to bicuspidata.

  1. bicuspidate var. bicuspidate has been changed to M. bicuspidata var. bicuspidata.

Line 215. Please change var. bicuspidate to bicuspidata.

  1. bicuspidate var. bicuspidate has been changed to M. bicuspidata var. bicuspidata.

Line 216. To complete the information Could the authors please indicate some yeast produced killer toxins against M. bicuspidata var. bicuspidate.

Ok. We have done this.

Line 219. Please change var. bicuspidate to bicuspidata

  1. bicuspidate var. bicuspidate has been changed to M. bicuspidata var. bicuspidata.

Line 221. Please change var. bicuspidate to bicuspidata.

  1. bicuspidate var. bicuspidate has been changed to M. bicuspidata var. bicuspidata.

Line 221. Please include M. bicuspidata.

Ok, We have done this.

Line 227. Please change var. bicuspidate to bicuspidata.

  1. bicuspidate var. bicuspidate has been changed to M. bicuspidata var. bicuspidata.

Line 230. Please change var. bicuspidate to bicuspidata.

  1. bicuspidate var. bicuspidate has been changed to M. bicuspidata var. bicuspidata.

Line 238. Please change var. bicuspidate to bicuspidata.

  1. bicuspidate var. bicuspidate has been changed to M. bicuspidata var. bicuspidata.

Line 240. Please change var. bicuspidate to bicuspidata.

  1. bicuspidate var. bicuspidate has been changed to M. bicuspidata var. bicuspidata.

Line 246. Please change var. bicuspidate to bicuspidata.

  1. bicuspidate var. bicuspidate has been changed to M. bicuspidata var. bicuspidata.

Line 252. Please change var. bicuspidate to bicuspidata.

  1. bicuspidate var. bicuspidate has been changed to M. bicuspidata var. bicuspidata.

Line 253. Use the adequate size typography in opportunistic.

Ok, we have done this.

Line 257. Please change var. bicuspidate to bicuspidata.

  1. bicuspidate var. bicuspidate has been changed to M. bicuspidata var. bicuspidata.

Line 259. Please change M. bicuspodate var. bicuspidate to M. bicuspidate var.

bicuspidata.

  1. bicuspidate var. bicuspidate has been changed to M. bicuspidata var. bicuspidata.

Line 264. Please change var. bicuspidate to bicuspidata.

  1. bicuspidate var. bicuspidate has been changed to M. bicuspidata var. bicuspidata.

Line 267. Please change var. bicuspidate to bicuspidata.

  1. bicuspidate var. bicuspidate has been changed to M. bicuspidata var. bicuspidata.

Line 274. Please change var. bicuspidate to bicuspidata.

  1. bicuspidate var. bicuspidate has been changed to M. bicuspidata var. bicuspidata.

Line 277. Please change var. bicuspidate to bicuspidata.

  1. bicuspidate var. bicuspidate has been changed to M. bicuspidata var. bicuspidata.

Line 280. Please change var. bicuspidate to bicuspidata.

  1. bicuspidate var. bicuspidate has been changed to M. bicuspidata var. bicuspidata.

Line 335. Please keep the capital letter in CRediT and lowercase letters after C.

Ok, we have done this.

Line 337. Please change M. bicuspidate var. bicuspidate to M. bicuspidate var.

bicuspidata.

  1. bicuspidate var. bicuspidate has been changed to M. bicuspidata var. bicuspidata.

Line 338. Please change ascospore yeast by ascosporogenous yeast.

ascospore yeast has been changed to ascosporogenous yeast.

Line 340. Please change M. bicuspidate var. bicuspidate to M. bicuspidate var.

bicuspidata.

  1. bicuspidate var. bicuspidate has been changed to M. bicuspidata var. bicuspidata.

Line 342. Please use the same size typography in symptoms) in different parts of

the dead body.

Ok, We have done these.

Line 349. Please change M. bicuspidate var. bicuspidate to M. bicuspidate var.

bicuspidata.

  1. bicuspidate var. bicuspidate has been changed to M. bicuspidata var. bicuspidata.

Line 267-399. Change Author, D. D., by Author, DD; as the authors guide indicates.

We have done this.

Line 267-399. Please put the abbreviated journal name in italic typography, year in bold letter as the authors guide indicate.

We have done this.
